# SAMPLE EFFICIENT ACTOR-CRITIC WITH EXPERIENCE REPLAY

**Ziyu Wang**
DeepMind
ziyu@google.com

**Victor Bapst**
DeepMind
vbapst@google.com

**Nicolas Heess**
DeepMind
heess@google.com

**Volodymyr Mnih**
DeepMind
vmnih@google.com

**Remi Munos**
DeepMind
Munos@google.com

**Koray Kavukcuoglu**
DeepMind
korayk@google.com

**Nando de Freitas**
DeepMind, CIFAR, Oxford University
nandodefreitas@google.com

## ABSTRACT

This paper presents an actor-critic deep reinforcement learning agent with experience replay that is stable, sample efficient, and performs remarkably well on challenging environments, including the discrete 57-game Atari domain and several continuous control problems. To achieve this, the paper introduces several innovations, including truncated importance sampling with bias correction, stochastic dueling network architectures, and a new trust region policy optimization method.

## 1 INTRODUCTION

Realistic simulated environments, where agents can be trained to learn a large repertoire of cognitive skills, are at the core of recent breakthroughs in AI (Bellemare et al., 2013; Mnih et al., 2015; Schulman et al., 2015a; Narasimhan et al., 2015; Mnih et al., 2016; Brockman et al., 2016; Oh et al., 2016). With richer realistic environments, the capabilities of our agents have increased and improved. Unfortunately, these advances have been accompanied by a substantial increase in the cost of simulation. In particular, every time an agent acts upon the environment, an expensive simulation step is conducted. Thus to reduce the cost of simulation, we need to reduce the number of simulation steps (i.e. samples of the environment). This need for sample efficiency is even more compelling when agents are deployed in the real world.

Experience replay (Lin, 1992) has gained popularity in deep $Q$-learning (Mnih et al., 2015; Schaul et al., 2016; Wang et al., 2016; Narasimhan et al., 2015), where it is often motivated as a technique for reducing sample correlation. Replay is actually a valuable tool for improving sample efficiency and, as we will see in our experiments, state-of-the-art deep $Q$-learning methods (Schaul et al., 2016; Wang et al., 2016) have been up to this point the most sample efficient techniques on Atari by a significant margin. However, we need to do better than deep $Q$-learning, because it has two important limitations. First, the deterministic nature of the optimal policy limits its use in adversarial domains. Second, finding the greedy action with respect to the $Q$ function is costly for large action spaces.

Policy gradient methods have been at the heart of significant advances in AI and robotics (Silver et al., 2014; Lillicrap et al., 2015; Silver et al., 2016; Levine et al., 2015; Mnih et al., 2016; Schulman et al., 2015a; Heess et al., 2015). Many of these methods are restricted to continuous domains or to very specific tasks such as playing Go. The existing variants applicable to both continuous and discrete domains, such as the on-policy asynchronous advantage actor critic (A3C) of Mnih et al. (2016), are sample inefficient.

The design of stable, sample efficient actor critic methods that apply to both continuous and discrete action spaces has been a long-standing hurdle of reinforcement learning (RL). We believe this paper

is the first to address this challenge successfully at scale. More specifically, we introduce an actor critic with experience replay (ACER) that nearly matches the state-of-the-art performance of deep $Q$-networks with prioritized replay on Atari, and substantially outperforms A3C in terms of sample efficiency on both Atari and continuous control domains.

ACER capitalizes on recent advances in deep neural networks, variance reduction techniques, the off-policy Retrace algorithm (Munos et al., 2016) and parallel training of RL agents (Mnih et al., 2016). Yet, crucially, its success hinges on innovations advanced in this paper: truncated importance sampling with bias correction, stochastic dueling network architectures, and efficient trust region policy optimization.

On the theoretical front, the paper proves that the Retrace operator can be rewritten from our proposed truncated importance sampling with bias correction technique.

## 2 BACKGROUND AND PROBLEM SETUP

Consider an agent interacting with its environment over discrete time steps. At time step $t$, the agent observes the $n_x$-dimensional state vector $x_t \in \mathcal{X} \subseteq \mathbb{R}^{n_x}$, chooses an action $a_t$ according to a policy $\pi(a|x_t)$ and observes a reward signal $r_t \in \mathbb{R}$ produced by the environment. We will consider discrete actions $a_t \in \{1, 2, \ldots, N_a\}$ in Sections 3 and 4, and continuous actions $a_t \in \mathcal{A} \subseteq \mathbb{R}^{n_a}$ in Section 5.

The goal of the agent is to maximize the discounted return $R_t = \sum_{i \geq 0} \gamma^i r_{t+i}$ in expectation. The discount factor $\gamma \in [0, 1)$ trades-off the importance of immediate and future rewards. For an agent following policy $\pi$, we use the standard definitions of the state-action and state only value functions:

$$Q^\pi(x_t, a_t) = \mathbb{E}_{x_{t+1:\infty}, a_{t+1:\infty}}\left[ R_t | x_t, a_t \right] \qquad \text{and} \qquad V^\pi(x_t) = \mathbb{E}_{a_t}\left[ Q^\pi(x_t, a_t) | x_t \right].$$

Here, the expectations are with respect to the observed environment states $x_t$ and the actions generated by the policy $\pi$, where $x_{t+1:\infty}$ denotes a state trajectory starting at time $t + 1$.

We also need to define the advantage function $A^\pi(x_t, a_t) = Q^\pi(x_t, a_t) - V^\pi(x_t)$, which provides a relative measure of value of each action since $\mathbb{E}_{a_t}\left[ A^\pi(x_t, a_t) \right] = 0$.

The parameters $\theta$ of the differentiable policy $\pi_\theta(a_t|x_t)$ can be updated using the discounted approximation to the policy gradient (Sutton et al., 2000), which borrowing notation from Schulman et al. (2015b), is defined as:

$$g = \mathbb{E}_{x_{0:\infty}, a_{0:\infty}}\left[ \sum_{t \geq 0} A^\pi(x_t, a_t) \nabla_\theta \log \pi_\theta(a_t|x_t) \right]. \tag{1}$$

Following Proposition 1 of Schulman et al. (2015b), we can replace $A^\pi(x_t, a_t)$ in the above expression with the state-action value $Q^\pi(x_t, a_t)$, the discounted return $R_t$, or the temporal difference residual $r_t + \gamma V^\pi(x_{t+1}) - V^\pi(x_t)$, without introducing bias. These choices will however have different variance. Moreover, in practice we will approximate these quantities with neural networks thus introducing additional approximation errors and biases. Typically, the policy gradient estimator using $R_t$ will have higher variance and lower bias whereas the estimators using function approximation will have higher bias and lower variance. Combining $R_t$ with the current value function approximation to minimize bias while maintaining bounded variance is one of the central design principles behind ACER.

To trade-off bias and variance, the asynchronous advantage actor critic (A3C) of Mnih et al. (2016) uses a single trajectory sample to obtain the following gradient approximation:

$$\widehat{g}^{\text{a3c}} = \sum_{t \geq 0} \left( \left( \sum_{i=0}^{k-1} \gamma^i r_{t+i} \right) + \gamma^k V_{\theta_v}^\pi(x_{t+k}) - V_{\theta_v}^\pi(x_t) \right) \nabla_\theta \log \pi_\theta(a_t|x_t). \tag{2}$$

A3C combines both $k$-step returns and function approximation to trade-off variance and bias. We may think of $V_{\theta_v}^\pi(x_t)$ as a policy gradient baseline used to reduce variance.

In the following section, we will introduce the discrete-action version of ACER. ACER may be understood as the off-policy counterpart of the A3C method of Mnih et al. (2016). As such, ACER builds on all the engineering innovations of A3C, including efficient parallel CPU computation.

ACER uses a single deep neural network to estimate the policy $\pi_\theta(a_t|x_t)$ and the value function $V_{\theta_v}^\pi(x_t)$. (For clarity and generality, we are using two different symbols to denote the parameters of the policy and value function, $\theta$ and $\theta_v$, but most of these parameters are shared in the single neural network.) Our neural networks, though building on the networks used in A3C, will introduce several modifications and new modules.

## 3 DISCRETE ACTOR CRITIC WITH EXPERIENCE REPLAY

Off-policy learning with experience replay may appear to be an obvious strategy for improving the sample efficiency of actor-critics. However, controlling the variance and stability of off-policy estimators is notoriously hard. Importance sampling is one of the most popular approaches for off-policy learning (Meuleau et al., 2000; Jie & Abbeel, 2010; Levine & Koltun, 2013). In our context, it proceeds as follows. Suppose we retrieve a trajectory $\{x_0, a_0, r_0, \mu(\cdot|x_0), \cdots, x_k, a_k, r_k, \mu(\cdot|x_k)\}$, where the actions have been sampled according to the behavior policy $\mu$, from our memory of experiences. Then, the importance weighted policy gradient is given by:

$$\widehat{g}^{\text{imp}} = \left(\prod_{t=0}^k \rho_t\right) \sum_{t=0}^k \left(\sum_{i=0}^k \gamma^i r_{t+i}\right) \nabla_\theta \log \pi_\theta(a_t|x_t), \tag{3}$$

where $\rho_t = \frac{\pi(a_t|x_t)}{\mu(a_t|x_t)}$ denotes the importance weight. This estimator is unbiased, but it suffers from very high variance as it involves a product of many potentially unbounded importance weights. To prevent the product of importance weights from exploding, Wawrzyński (2009) truncates this product. Truncated importance sampling over entire trajectories, although bounded in variance, could suffer from significant bias.

Recently, Degris et al. (2012) attacked this problem by using marginal value functions over the limiting distribution of the process to yield the following approximation of the gradient:

$$g^{\text{marg}} = \mathbb{E}_{x_t \sim \beta, a_t \sim \mu}\left[\rho_t \nabla_\theta \log \pi_\theta(a_t|x_t) Q^\pi(x_t, a_t)\right], \tag{4}$$

where $\mathbb{E}_{x_t \sim \beta, a_t \sim \mu}[\cdot]$ is the expectation with respect to the limiting distribution $\beta(x) = \lim_{t \to \infty} P(x_t = x|x_0, \mu)$ with behavior policy $\mu$. To keep the notation succinct, we will replace $\mathbb{E}_{x_t \sim \beta, a_t \sim \mu}[\cdot]$ with $\mathbb{E}_{x_t a_t}[\cdot]$ and ensure we remind readers of this when necessary.

Two important facts about equation (4) must be highlighted. First, note that it depends on $Q^\pi$ and not on $Q^\mu$, consequently we must be able to estimate $Q^\pi$. Second, we no longer have a product of importance weights, but instead only need to estimate the marginal importance weight $\rho_t$. Importance sampling in this lower dimensional space (over marginals as opposed to trajectories) is expected to exhibit lower variance.

Degris et al. (2012) estimate $Q^\pi$ in equation (4) using *lambda returns*: $R_t^\lambda = r_t + (1-\lambda)\gamma V(x_{t+1}) + \lambda\gamma\rho_{t+1}R_{t+1}^\lambda$. This estimator requires that we know how to choose $\lambda$ ahead of time to trade off bias and variance. Moreover, when using small values of $\lambda$ to reduce variance, occasional large importance weights can still cause instability.

In the following subsection, we adopt the Retrace algorithm of Munos et al. (2016) to estimate $Q^\pi$. Subsequently, we propose an importance weight truncation technique to improve the stability of the off-policy actor critic of Degris et al. (2012), and introduce a computationally efficient trust region scheme for policy optimization. The formulation of ACER for continuous action spaces will require further innovations that are advanced in Section 5.

### 3.1 MULTI-STEP ESTIMATION OF THE STATE-ACTION VALUE FUNCTION

In this paper, we estimate $Q^\pi(x_t, a_t)$ using Retrace (Munos et al., 2016). (We also experimented with the related tree backup method of Precup et al. (2000) but found Retrace to perform better in practice.) Given a trajectory generated under the behavior policy $\mu$, the Retrace estimator can be expressed recursively as follows[1]:

$$Q^{\text{ret}}(x_t, a_t) = r_t + \gamma\bar{\rho}_{t+1}[Q^{\text{ret}}(x_{t+1}, a_{t+1}) - Q(x_{t+1}, a_{t+1})] + \gamma V(x_{t+1}), \tag{5}$$

---

[1]For ease of presentation, we consider only $\lambda = 1$ for Retrace.

where $\bar{\rho}_t$ is the truncated importance weight, $\bar{\rho}_t = \min\{c, \rho_t\}$ with $\rho_t = \frac{\pi(a_t|x_t)}{\mu(a_t|x_t)}$, $Q$ is the current value estimate of $Q^\pi$, and $V(x) = \mathbb{E}_{a\sim\pi}Q(x, a)$. Retrace is an off-policy, return-based algorithm which has low variance and is proven to converge (in the tabular case) to the value function of the target policy for any behavior policy, see Munos et al. (2016).

The recursive Retrace equation depends on the estimate $Q$. To compute it, in discrete action spaces, we adopt a convolutional neural network with "two heads" that outputs the estimate $Q_{\theta_v}(x_t, a_t)$, as well as the policy $\pi_\theta(a_t|x_t)$. This neural representation is the same as in (Mnih et al., 2016), with the exception that we output the vector $Q_{\theta_v}(x_t, a_t)$ instead of the scalar $V_{\theta_v}(x_t)$. The estimate $V_{\theta_v}(x_t)$ can be easily derived by taking the expectation of $Q_{\theta_v}$ under $\pi_\theta$.

To approximate the policy gradient $g^{\text{marg}}$, ACER uses $Q^{\text{ret}}$ to estimate $Q^\pi$. As Retrace uses multi-step returns, it can significantly reduce bias in the estimation of the policy gradient [2].

To learn the critic $Q_{\theta_v}(x_t, a_t)$, we again use $Q^{\text{ret}}(x_t, a_t)$ as a target in a mean squared error loss and update its parameters $\theta_v$ with the following standard gradient:

$$(Q^{\text{ret}}(x_t, a_t) - Q_{\theta_v}(x_t, a_t))\nabla_{\theta_v}Q_{\theta_v}(x_t, a_t). \tag{6}$$

Because Retrace is return-based, it also enables faster learning of the critic. Thus the purpose of the multi-step estimator $Q^{\text{ret}}$ in our setting is twofold: to reduce bias in the policy gradient, and to enable faster learning of the critic, hence further reducing bias.

### 3.2 Importance Weight Truncation with Bias Correction

The marginal importance weights in Equation (4) can become large, thus causing instability. To safe-guard against high variance, we propose to truncate the importance weights and introduce a correction term via the following decomposition of $g^{\text{marg}}$:

$$g^{\text{marg}} = \mathbb{E}_{x_t a_t}\left[\rho_t \nabla_\theta \log \pi_\theta(a_t|x_t) Q^\pi(x_t, a_t)\right]$$

$$= \mathbb{E}_{x_t}\left[\mathbb{E}_{a_t}[\bar{\rho}_t \nabla_\theta \log \pi_\theta(a_t|x_t) Q^\pi(x_t, a_t)] + \underset{a\sim\pi}{\mathbb{E}}\left(\left[\frac{\rho_t(a)-c}{\rho_t(a)}\right]_+ \nabla_\theta \log \pi_\theta(a|x_t) Q^\pi(x_t, a)\right)\right], \tag{7}$$

where $\bar{\rho}_t = \min\{c, \rho_t\}$ with $\rho_t = \frac{\pi(a_t|x_t)}{\mu(a_t|x_t)}$ as before. We have also introduced the notation $\rho_t(a) = \frac{\pi(a|x_t)}{\mu(a|x_t)}$, and $[x]_+ = x$ if $x > 0$ and it is zero otherwise. We remind readers that the above expectations are with respect to the limiting state distribution under the behavior policy: $x_t \sim \beta$ and $a_t \sim \mu$.

The clipping of the importance weight in the first term of equation (7) ensures that the variance of the gradient estimate is bounded. The correction term (second term in equation (7)) ensures that our estimate is *unbiased*. Note that the correction term is only active for actions such that $\rho_t(a) > c$. In particular, if we choose a large value for $c$, the correction term only comes into effect when the variance of the original off-policy estimator of equation (4) is very high. When this happens, our decomposition has the nice property that the truncated weight in the first term is at most $c$ while the correction weight $\left[\frac{\rho_t(a)-c}{\rho_t(a)}\right]_+$ in the second term is at most 1.

We model $Q^\pi(x_t, a)$ in the correction term with our neural network approximation $Q_{\theta_v}(x_t, a_t)$. This modification results in what we call the *truncation with bias correction trick*, in this case applied to the function $\nabla_\theta \log \pi_\theta(a_t|x_t) Q^\pi(x_t, a_t)$:

$$\widehat{g}^{\text{marg}} = \mathbb{E}_{x_t}\left[\mathbb{E}_{a_t}[\bar{\rho}_t \nabla_\theta \log \pi_\theta(a_t|x_t) Q^{ret}(x_t, a_t)] + \underset{a\sim\pi}{\mathbb{E}}\left(\left[\frac{\rho_t(a)-c}{\rho_t(a)}\right]_+ \nabla_\theta \log \pi_\theta(a|x_t) Q_{\theta_v}(x_t, a)\right)\right]. \tag{8}$$

Equation (8) involves an expectation over the stationary distribution of the Markov process. We can however approximate it by sampling trajectories $\{x_0, a_0, r_0, \mu(\cdot|x_0), \cdots, x_k, a_k, r_k, \mu(\cdot|x_k)\}$

---

[2] An alternative to Retrace here is $Q(\lambda)$ with off-policy corrections (Harutyunyan et al., 2016) which we discuss in more detail in Appendix B.

generated from the behavior policy $\mu$. Here the terms $\mu(\cdot|x_t)$ are the policy vectors. Given these trajectories, we can compute the off-policy ACER gradient:

$$
\begin{aligned}
\widehat{g}_t^{\text{acer}} \quad = \quad & \bar{\rho}_t \nabla_\theta \log \pi_\theta(a_t|x_t)[Q^{\text{ret}}(x_t, a_t) - V_{\theta_v}(x_t)] \\
& + \underset{a\sim\pi}{\mathbb{E}} \left( \left[ \frac{\rho_t(a) - c}{\rho_t(a)} \right]_+ \nabla_\theta \log \pi_\theta(a|x_t)[Q_{\theta_v}(x_t, a) - V_{\theta_v}(x_t)] \right).
\end{aligned}
\tag{9}
$$

In the above expression, we have subtracted the classical baseline $V_{\theta_v}(x_t)$ to reduce variance.

It is interesting to note that, when $c = \infty$, (9) recovers (off-policy) policy gradient up to the use of Retrace. When $c = 0$, (9) recovers an actor critic update that depends entirely on $Q$ estimates. In the continuous control domain, (9) also generalizes Stochastic Value Gradients if $c = 0$ and the reparametrization trick is used to estimate its second term (Heess et al., 2015).

### 3.3 EFFICIENT TRUST REGION POLICY OPTIMIZATION

The policy updates of actor-critic methods do often exhibit high variance. Hence, to ensure stability, we must limit the per-step changes to the policy. Simply using smaller learning rates is insufficient as they cannot guard against the occasional large updates while maintaining a desired learning speed. Trust Region Policy Optimization (TRPO) (Schulman et al., 2015a) provides a more adequate solution.

Schulman et al. (2015a) approximately limit the difference between the updated policy and the current policy to ensure safety. Despite the effectiveness of their TRPO method, it requires repeated computation of Fisher-vector products for each update. This can prove to be prohibitively expensive in large domains.

In this section we introduce a new trust region policy optimization method that scales well to large problems. Instead of constraining the updated policy to be close to the current policy (as in TRPO), we propose to maintain an *average policy network* that represents a running average of past policies and forces the updated policy to not deviate far from this average.

We decompose our policy network in two parts: a distribution $f$, and a deep neural network that generates the statistics $\phi_\theta(x)$ of this distribution. That is, given $f$, the policy is completely characterized by the network $\phi_\theta$: $\pi(\cdot|x) = f(\cdot|\phi_\theta(x))$. For example, in the discrete domain, we choose $f$ to be the categorical distribution with a probability vector $\phi_\theta(x)$ as its statistics. The probability vector is of course parameterised by $\theta$.

We denote the average policy network as $\phi_{\theta_a}$ and update its parameters $\theta_a$ "softly" after each update to the policy parameter $\theta$: $\theta_a \leftarrow \alpha\theta_a + (1 - \alpha)\theta$.

Consider, for example, the ACER policy gradient as defined in Equation (9), *but with respect to $\phi$:*

$$
\begin{aligned}
\widehat{g}_t^{\text{acer}} \quad = \quad & \bar{\rho}_t \nabla_{\phi_\theta(x)} \log f(a_t|\phi_\theta(x))[Q^{\text{ret}}(x_t, a_t) - V_{\theta_v}(x_t)] \\
& + \underset{a\sim\pi}{\mathbb{E}} \left( \left[ \frac{\rho_t(a) - c}{\rho_t(a)} \right]_+ \nabla_{\phi_\theta(x)} \log f(a_t|\phi_\theta(x))[Q_{\theta_v}(x_t, a) - V_{\theta_v}(x_t)] \right).
\end{aligned}
\tag{10}
$$

Given the averaged policy network, our proposed trust region update involves two stages. In the first stage, we solve the following optimization problem with a linearized KL divergence constraint:

$$
\begin{aligned}
\underset{z}{\text{minimize}} \quad & \frac{1}{2}\|\widehat{g}_t^{\text{acer}} - z\|_2^2 \\
\text{subject to} \quad & \nabla_{\phi_\theta(x_t)} D_{KL}\left[ f(\cdot|\phi_{\theta_a}(x_t)) \| f(\cdot|\phi_\theta(x_t)) \right]^T z \leq \delta
\end{aligned}
\tag{11}
$$

Since the constraint is linear, the overall optimization problem reduces to a simple quadratic programming problem, the solution of which can be easily derived in closed form using the KKT conditions. Letting $k = \nabla_{\phi_\theta(x_t)} D_{KL}\left[ f(\cdot|\phi_{\theta_a}(x_t) \| f(\cdot|\phi_\theta(x_t)) \right]$, the solution is:

$$
z^* = \widehat{g}_t^{\text{acer}} - \max \left\{ 0, \frac{k^T \widehat{g}_t^{\text{acer}} - \delta}{\|k\|_2^2} \right\} k
\tag{12}
$$

This transformation of the gradient has a very natural form. If the constraint is satisfied, there is no change to the gradient with respect to $\phi_\theta(x_t)$. Otherwise, the update is scaled down in the direction

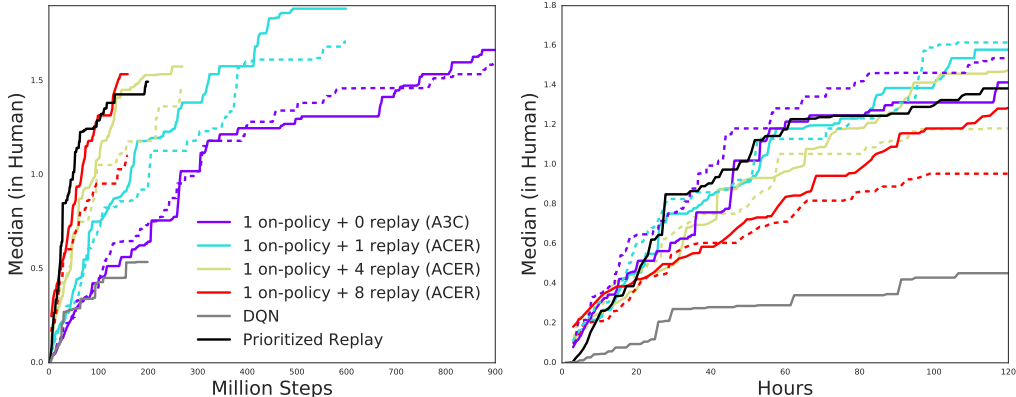

Figure 1: ACER improvements in sample (**LEFT**) and computation (**RIGHT**) complexity on Atari. On each plot, the median of the human-normalized score across all 57 Atari games is presented for 4 ratios of replay with 0 replay corresponding to on-policy A3C. The colored solid and dashed lines represent ACER with and without trust region updating respectively. The environment steps are counted over all threads. The gray curve is the original DQN agent (Mnih et al., 2015) and the black curve is one of the Prioritized Double DQN agents from Schaul et al. (2016).

of $k$, thus effectively lowering rate of change between the activations of the current policy and the average policy network.

In the second stage, we take advantage of back-propagation. Specifically, the updated gradient with respect to $\phi_\theta$, that is $z^*$, is back-propagated through the network to compute the derivatives with respect to the parameters. The parameter updates for the policy network follow from the chain rule: $\frac{\partial \phi_\theta(x)}{\partial \theta} z^*$.

The trust region step is carried out in the space of the statistics of the distribution $f$, and not in the space of the policy parameters. This is done deliberately so as to avoid an additional back-propagation step through the policy network.

We would like to remark that the algorithm advanced in this section can be thought of as a general strategy for modifying the backward messages in back-propagation so as to stabilize the activations.

Instead of a trust region update, one could alternatively add an appropriately scaled KL cost to the objective function as proposed by Heess et al. (2015). This approach, however, is less robust to the choice of hyper-parameters in our experience.

The ACER algorithm results from a combination of the above ideas, with the precise pseudo-code appearing in Appendix A. A master algorithm (Algorithm 1) calls ACER on-policy to perform updates and propose trajectories. It then calls ACER off-policy component to conduct several replay steps. When on-policy, ACER effectively becomes a modified version of A3C where $Q$ instead of $V$ baselines are employed and trust region optimization is used.

## 4 RESULTS ON ATARI

We use the Arcade Learning Environment of Bellemare et al. (2013) to conduct an extensive evaluation. We deploy one single algorithm and network architecture, with fixed hyper-parameters, to learn to play 57 Atari games given only raw pixel observations and game rewards. This task is highly demanding because of the diversity of games, and high-dimensional pixel-level observations.

Our experimental setup uses 16 actor-learner threads running on a single machine with no GPUs. We adopt the same input pre-processing and network architecture as Mnih et al. (2015). Specifically, the network consists of a convolutional layer with 32 $8 \times 8$ filters with stride 4 followed by another convolutional layer with 64 $4 \times 4$ filters with stride 2, followed by a final convolutional layer with 64 $3 \times 3$ filters with stride 1, followed by a fully-connected layer of size 512. Each of the hidden layers is followed by a rectifier nonlinearity. The network outputs a softmax policy and $Q$ values.

When using replay, we add to each thread a replay memory that is up to $50\,000$ frames in size. The total amount of memory used across all threads is thus similar in size to that of DQN (Mnih et al., 2015). For all Atari experiments, we use a single learning rate adopted from an earlier implementation of A3C without further tuning. We do not anneal the learning rates over the course of training as in Mnih et al. (2016). We otherwise adopt the same optimization procedure as in Mnih et al. (2016). Specifically, we adopt entropy regularization with weight $0.001$, discount the rewards with $\gamma = 0.99$, and perform updates every 20 steps ($k = 20$ in the notation of Section 2). In all our experiments with experience replay, we use importance weight truncation with $c = 10$. We consider training ACER both with and without trust region updating as described in Section 3.3. When trust region updating is used, we use $\delta = 1$ and $\alpha = 0.99$ for all experiments.

To compare different agents, we adopt as our metric the median of the human normalized score over all 57 games. The normalization is calculated such that, for each game, human scores and random scores are evaluated to 1, and 0 respectively. The normalized score for a given game at time $t$ is computed as the average normalized score over the past 1 million consecutive frames encountered until time $t$. For each agent, we plot its cumulative maximum median score over time. The result is summarized in Figure 1.

The four colors in Figure 1 correspond to four replay ratios (0, 1, 4 and 8) with a ratio of 4 meaning that we use the off-policy component of ACER 4 times after using the on-policy component (A3C). That is, a replay ratio of 0 means that we are using A3C. The solid and dashed lines represent ACER with and without trust region updating respectively. The gray and black curves are the original DQN (Mnih et al., 2015) and Prioritized Replay agent of Schaul et al. (2016) agents respectively.

As shown on the left panel of Figure 1, replay significantly increases data efficiency. We observe that when using the trust region optimizer, the average reward as a function of the number of environmental steps increases with the ratio of replay. This increase has diminishing returns, but with enough replay, ACER can match the performance of the best DQN agents. Moreover, it is clear that the off-policy actor critics (ACER) are much more sample efficient than their on-policy counterpart (A3C).

The right panel of Figure 1 shows that ACER agents perform similarly to A3C when measured by wall clock time. Thus, in this case, it is possible to achieve better data-efficiency without necessarily compromising on computation time. In particular, ACER with a replay ratio of 4 is an appealing alternative to either the prioritized DQN agent or A3C.

## 5 CONTINUOUS ACTOR CRITIC WITH EXPERIENCE REPLAY

Retrace requires estimates of both $Q$ and $V$, but we cannot easily integrate over $Q$ to derive $V$ in continuous action spaces. In this section, we propose a solution to this problem in the form of a novel representation for RL, as well as modifications necessary for trust region updating.

### 5.1 POLICY EVALUATION

Retrace provides a target for learning $Q_{\theta_v}$, but not for learning $V_{\theta_v}$. We could use importance sampling to compute $V_{\theta_v}$ given $Q_{\theta_v}$, but this estimator has high variance.

We propose a new architecture which we call Stochastic Dueling Networks (SDNs), inspired by the Dueling networks of Wang et al. (2016), which is designed to estimate both $V^\pi$ and $Q^\pi$ off-policy while maintaining consistency between the two estimates. At each time step, an SDN outputs a stochastic estimate $\widetilde{Q}_{\theta_v}$ of $Q^\pi$ and a deterministic estimate $V_{\theta_v}$ of $V^\pi$, such that

$$\widetilde{Q}_{\theta_v}(x_t, a_t) \sim V_{\theta_v}(x_t) + A_{\theta_v}(x_t, a_t) - \frac{1}{n}\sum_{i=1}^{n} A_{\theta_v}(x_t, u_i), \text{ and } u_i \sim \pi_\theta(\cdot|x_t) \qquad (13)$$

where $n$ is a parameter, see Figure 2. The two estimates are consistent in the sense that $\mathbb{E}_{a\sim\pi(\cdot|x_t)}\left[\mathbb{E}_{u_{1:n}\sim\pi(\cdot|x_t)}\left(\widetilde{Q}_{\theta_v}(x_t, a)\right)\right] = V_{\theta_v}(x_t)$. Furthermore, we can learn about $V^\pi$ by learning $\widetilde{Q}_{\theta_v}$. To see this, assume we have learned $Q^\pi$ perfectly such that $\mathbb{E}_{u_{1:n}\sim\pi(\cdot|x_t)}\left(\widetilde{Q}_{\theta_v}(x_t, a_t)\right) = Q^\pi(x_t, a_t)$, then $V_{\theta_v}(x_t) = \mathbb{E}_{a\sim\pi(\cdot|x_t)}\left[\mathbb{E}_{u_{1:n}\sim\pi(\cdot|x_t)}\left(\widetilde{Q}_{\theta_v}(x_t, a)\right)\right] = \mathbb{E}_{a\sim\pi(\cdot|x_t)}[Q^\pi(x_t, a)] = V^\pi(x_t)$. Therefore, a target on $\widetilde{Q}_{\theta_v}(x_t, a_t)$ also provides an error signal for updating $V_{\theta_v}$.

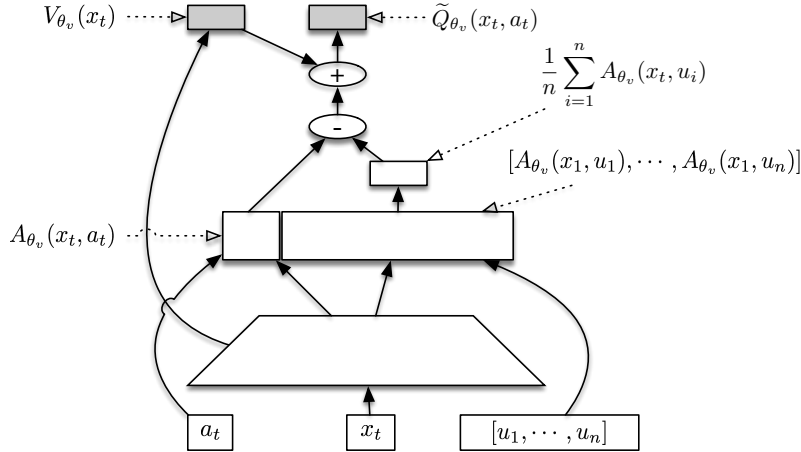

Figure 2: A schematic of the Stochastic Dueling Network. In the drawing, $[u_1, \cdots, u_n]$ are assumed to be samples from $\pi_\theta(\cdot|x_t)$. This schematic illustrates the concept of SDNs but does not reflect the real sizes of the networks used.

In addition to SDNs, however, we also construct the following novel target for estimating $V^\pi$:

$$V^{target}(x_t) = \min\left\{1, \frac{\pi(a_t|x_t)}{\mu(a_t|x_t)}\right\}\left(Q^{\text{ret}}(x_t, a_t) - Q_{\theta_v}(x_t, a_t)\right) + V_{\theta_v}(x_t). \qquad (14)$$

The above target is also derived via the truncation and bias correction trick; for more details, see Appendix D.

Finally, when estimating $Q^{\text{ret}}$ in continuous domains, we implement a slightly different formulation of the truncated importance weights $\bar{\rho}_t = \min\left\{1, \left(\frac{\pi(a_t|x_t)}{\mu(a_t|x_t)}\right)^{\frac{1}{d}}\right\}$, where $d$ is the dimensionality of the action space. Although not essential, we have found this formulation to lead to faster learning.

## 5.2 TRUST REGION UPDATING

To adopt the trust region updating scheme (Section 3.3) in the continuous control domain, one simply has to choose a distribution $f$ and a gradient specification $\hat{g}_t^{\text{acer}}$ suitable for continuous action spaces.

For the distribution $f$, we choose Gaussian distributions with fixed diagonal covariance and mean $\phi_\theta(x)$.

To derive $\hat{g}_t^{\text{acer}}$ in continuous action spaces, consider the ACER policy gradient for the stochastic dueling network, but with respect to $\phi$:

$$\begin{aligned}
g_t^{\text{acer}} &= \mathbb{E}_{x_t}\left[\mathbb{E}_{a_t}\left[\bar{\rho}_t \nabla_{\phi_\theta(x_t)} \log f(a_t|\phi_\theta(x_t))(Q^{\text{opc}}(x_t, a_t) - V_{\theta_v}(x_t))\right]\right.\\
&\left.+ \underset{a \sim \pi}{\mathbb{E}}\left(\left[\frac{\rho_t(a) - c}{\rho_t(a)}\right]_+ (\widetilde{Q}_{\theta_v}(x_t, a) - V_{\theta_v}(x_t))\nabla_{\phi_\theta(x_t)} \log f(a|\phi_\theta(x_t))\right)\right]. \qquad (15)
\end{aligned}$$

In the above definition, we are using $Q^{\text{opc}}$ instead of $Q^{\text{ret}}$. Here, $Q^{\text{opc}}(x_t, a_t)$ is the same as Retrace with the exception that the truncated importance ratio is replaced with 1 (Harutyunyan et al., 2016). Please refer to Appendix B an expanded discussion on this design choice. Given an observation $x_t$, we can sample $a'_t \sim \pi_\theta(\cdot|x_t)$ to obtain the following Monte Carlo approximation

$$\begin{aligned}
\hat{g}_t^{\text{acer}} &= \bar{\rho}_t \nabla_{\phi_\theta(x_t)} \log f(a_t|\phi_\theta(x_t))(Q^{\text{opc}}(x_t, a_t) - V_{\theta_v}(x_t))\\
&+ \left[\frac{\rho_t(a'_t) - c}{\rho_t(a'_t)}\right]_+ (\widetilde{Q}_{\theta_v}(x_t, a'_t) - V_{\theta_v}(x_t))\nabla_{\phi_\theta(x_t)} \log f(a'_t|\phi_\theta(x_t)). \qquad (16)
\end{aligned}$$

Given $f$ and $\hat{g}_t^{\text{acer}}$, we apply the same steps as detailed in Section 3.3 to complete the update.

The precise pseudo-code of ACER algorithm for continuous spaces results is presented in Appendix A.

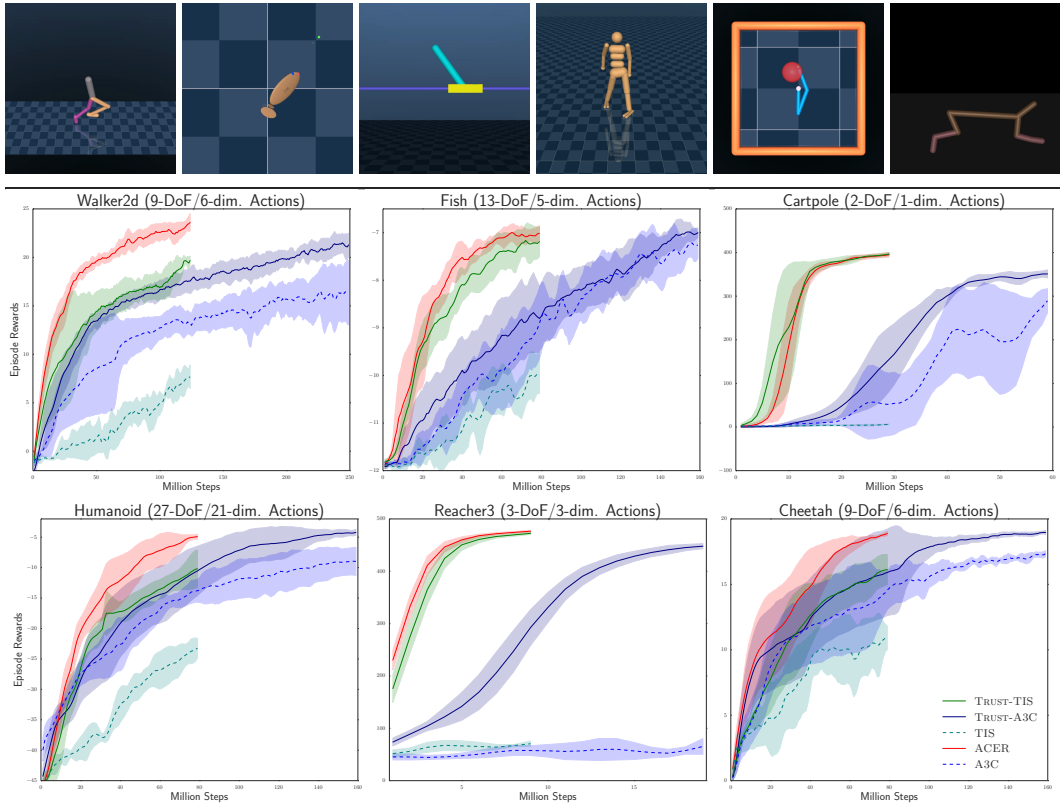

Figure 3: **[TOP]** Screen shots of the continuous control tasks. **[BOTTOM]** Performance of different methods on these tasks. ACER outperforms all other methods and shows clear gains for the higher-dimensionality tasks (humanoid, cheetah, walker and fish). The proposed trust region method by itself improves the two baselines (truncated importance sampling and A3C) significantly.

# 6    RESULTS ON MUJOCO

We evaluate our algorithms on 6 continuous control tasks, all of which are simulated using the MuJoCo physics engine (Todorov et al., 2012). For descriptions of the tasks, please refer to Appendix E.1. Briefly, the tasks with action dimensionality in brackets are: cartpole (1D), reacher (3D), cheetah (6D), fish (5D), walker (6D) and humanoid (21D). These tasks are illustrated in Figure 3.

To benchmark ACER for continuous control, we compare it to its on-policy counterpart both with and without trust region updating. We refer to these two baselines as A3C and Trust-A3C. Additionally, we also compare to a baseline with replay where we truncate the importance weights over trajectories as in (Wawrzyński, 2009). For a detailed description of this baseline, please refer to Appendix E. Again, we run this baseline both with and without trust region updating, and refer to these choices as Trust-TIS and TIS respectively. Last but not least, we refer to our proposed approach with SDN and trust region updating as simply ACER. All five setups are implemented in the asynchronous A3C framework.

All the aforementioned setups share the same network architecture that computes the policy and state values. We maintain an additional small network that computes the stochastic $A$ values in the case of ACER. We use $n = 5$ (using the notation in Equation (13)) in all SDNs. Instead of mixing on-policy and replay learning as done in the Atari domain, ACER for continuous actions is entirely off-policy, with experiences generated from the simulator (4 times on average). When using replay, we add to each thread a replay memory that is $5,000$ frames in size and perform updates every 50 steps ($k = 50$ in the notation of Section 2). The rate of the soft updating ($\alpha$ as in Section 3.3) is set to $0.995$ in all setups involving trust region updating. The truncation threshold $c$ is set to $5$ for ACER.

We use diagonal Gaussian policies with fixed diagonal covariances where the diagonal standard deviation is set to $0.3$. For all setups, we sample the learning rates log-uniformly in the range $[10^{-4}, 10^{-3.3}]$. For setups involving trust region updating, we also sample $\delta$ uniformly in the range $[0.1, 2]$. With all setups, we use 30 sampled hyper-parameter settings.

The empirical results for all continuous control tasks are shown Figure 3, where we show the mean and standard deviation of the best 5 out of 30 hyper-parameter settings over which we searched [3]. For sensitivity analyses with respect to the hyper-parameters, please refer to Figures 5 and 6 in the Appendix.

In continuous control, ACER outperforms the A3C and truncated importance sampling baselines by a very significant margin.

Here, we also find that the proposed trust region optimization method can result in huge improvements over the baselines. The high-dimensional continuous action policies are much harder to optimize than the small discrete action policies in Atari, and hence we observe much higher gains for trust region optimization in the continuous control domains. In spite of the improvements brought in by trust region optimization, ACER still outperforms all other methods, specially in higher dimensions.

### 6.1 ABLATIONS

To further tease apart the contributions of the different components of ACER, we conduct an ablation analysis where we individually remove Retrace / Q($\lambda$) off-policy correction, SDNs, trust region, and truncation with bias correction from the algorithm. As shown in Figure 4, Retrace and off-policy correction, SDNs, and trust region are critical: removing any one of them leads to a clear deterioration of the performance. Truncation with bias correction did not alter the results in the Fish and Walker2d tasks. However, in Humanoid, where the dimensionality of the action space is much higher, including truncation and bias correction brings a significant boost which makes the originally kneeling humanoid stand. Presumably, the high dimensionality of the action space increases the variance of the importance weights which makes truncation with bias correction important. For more details on the experimental setup please see Appendix E.4.

## 7 THEORETICAL ANALYSIS

Retrace is a very recent development in reinforcement learning. In fact, this work is the first to consider Retrace in the policy gradients setting. For this reason, and given the core role that Retrace plays in ACER, it is valuable to shed more light on this technique. In this section, we will prove that Retrace can be interpreted as an application of the importance weight truncation and bias correction trick advanced in this paper.

Consider the following equation:

$$Q^{\pi}(x_t, a_t) = \mathbb{E}_{x_{t+1}a_{t+1}} \left[ r_t + \gamma \rho_{t+1} Q^{\pi}(x_{t+1}, a_{t+1}) \right]. \tag{17}$$

If we apply the weight truncation and bias correction trick to the above equation we obtain

$$Q^{\pi}(x_t, a_t) = \mathbb{E}_{x_{t+1}a_{t+1}} \left[ r_t + \gamma \bar{\rho}_{t+1} Q^{\pi}(x_{t+1}, a_{t+1}) + \gamma \mathbb{E}_{a \sim \pi} \left( \left[ \frac{\rho_{t+1}(a) - c}{\rho_{t+1}(a)} \right]_+ Q^{\pi}(x_{t+1}, a) \right) \right]. \tag{18}$$

By recursively expanding $Q^{\pi}$ as in Equation (18), we can represent $Q^{\pi}(x, a)$ as:

$$Q^{\pi}(x, a) = \mathbb{E}_{\mu} \left[ \sum_{t \geq 0} \gamma^t \left( \prod_{i=1}^{t} \bar{\rho}_i \right) \left( r_t + \gamma \mathbb{E}_{b \sim \pi} \left( \left[ \frac{\rho_{t+1}(b) - c}{\rho_{t+1}(b)} \right]_+ Q^{\pi}(x_{t+1}, b) \right) \right) \right]. \tag{19}$$

The expectation $\mathbb{E}_{\mu}$ is taken over trajectories starting from $x$ with actions generated with respect to $\mu$. When $Q^{\pi}$ is not available, we can replace it with our current estimate $Q$ to get a return-based

---

[3] For videos of the policies learned with ACER, please see: https://www.youtube.com/watch?v=NmbeQYoVv5g&list=PLkmHIkhlFjiTlvwxEnsJMs3v7seR5HSP-.

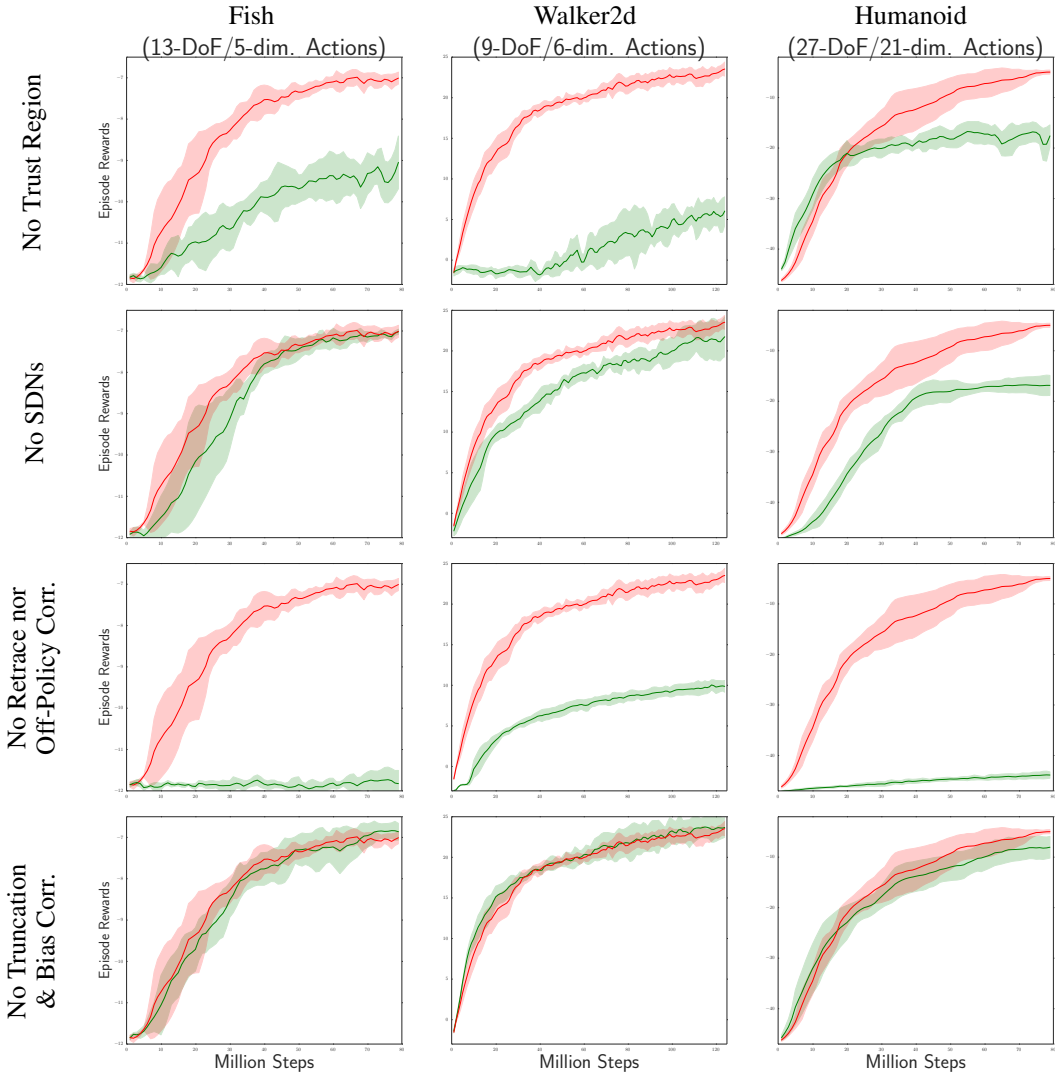

Figure 4: Ablation analysis evaluating the effect of different components of ACER. Each row compares ACER with and without one component. The columns represents three control tasks. Red lines, in all plots, represent ACER whereas green lines ACER with missing components. This study indicates that all 4 components studied improve performance where 3 are critical to success. Note that the ACER curve is of course the same in all rows.

esitmate of $Q^\pi$. This operation also defines an operator:

$$\mathcal{B}Q(x,a) = \mathbb{E}_\mu \left[ \sum_{t\geq 0} \gamma^t \left( \prod_{i=1}^t \bar{\rho}_i \right) \left( r_t + \gamma \mathbb{E}_{b\sim\pi} \left( \left[ \frac{\rho_{t+1}(b) - c}{\rho_{t+1}(b)} \right]_+ Q(x_{t+1}, b) \right) \right) \right]. \quad (20)$$

In the following proposition, we show that $\mathcal{B}$ is a contraction operator with a unique fixed point $Q^\pi$ and that it is equivalent to the Retrace operator.

**Proposition 1.** *The operator $\mathcal{B}$ is a contraction operator such that $\|\mathcal{B}Q - Q^\pi\|_\infty \leq \gamma\|Q - Q^\pi\|_\infty$ and $\mathcal{B}$ is equivalent to Retrace.*

The above proposition not only shows an alternative way of arriving at the same operator, but also provides a different proof of contraction for Retrace. Please refer to Appendix C for the regularization conditions and proof of the above proposition.

Finally, $\mathcal{B}$, and therefore Retrace, generalizes both the Bellman operator $\mathcal{T}^\pi$ and importance sampling. Specifically, when $c = 0$, $\mathcal{B} = \mathcal{T}^\pi$ and when $c = \infty$, $\mathcal{B}$ recovers importance sampling; see Appendix C.

# 8 CONCLUDING REMARKS

We have introduced a stable off-policy actor critic that scales to both continuous and discrete action spaces. This approach integrates several recent advances in RL in a principle manner. In addition, it integrates three innovations advanced in this paper: truncated importance sampling with bias correction, stochastic dueling networks and an efficient trust region policy optimization method.

We showed that the method not only matches the performance of the best known methods on Atari, but that it also outperforms popular techniques on several continuous control problems.

The efficient trust region optimization method advanced in this paper performs remarkably well in continuous domains. It could prove very useful in other deep learning domains, where it is hard to stabilize the training process.

## ACKNOWLEDGMENTS

We are very thankful to Marc Bellemare, Jascha Sohl-Dickstein, and Sébastien Racaniere for proof-reading and valuable suggestions.

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

# A   ACER PSEUDO-CODE FOR DISCRETE ACTIONS

---

**Algorithm 1** ACER for discrete actions (master algorithm)

---

// *Assume global shared parameter vectors $\theta$ and $\theta_v$.*
// *Assume ratio of replay $r$.*
**repeat**
 Call ACER on-policy, Algorithm 2.
 $n \leftarrow \text{Possion}(r)$
 **for** $i \in \{1, \cdots, n\}$ **do**
 Call ACER off-policy, Algorithm 2.
 **end for**
**until** Max iteration or time reached.

---

**Algorithm 2** ACER for discrete actions

---

Reset gradients $d\theta \leftarrow 0$ and $d\theta_v \leftarrow 0$.
Initialize parameters $\theta' \leftarrow \theta$ and $\theta'_v \leftarrow \theta_v$.
**if not** On-Policy **then**
 Sample the trajectory $\{x_0, a_0, r_0, \mu(\cdot|x_0), \cdots, x_k, a_k, r_k, \mu(\cdot|x_k)\}$ from the replay memory.
**else**
 Get state $x_0$
**end if**
**for** $i \in \{0, \cdots, k\}$ **do**
 Compute $f(\cdot|\phi_{\theta'}(x_i))$, $Q_{\theta'_v}(x_i, \cdot)$ and $f(\cdot|\phi_{\theta_a}(x_i))$.
 **if** On-Policy **then**
 Perform $a_i$ according to $f(\cdot|\phi_{\theta'}(x_i))$
 Receive reward $r_i$ and new state $x_{i+1}$
 $\mu(\cdot|x_i) \leftarrow f(\cdot|\phi_{\theta'}(x_i))$
 **end if**
 $\bar{\rho}_i \leftarrow \min\left\{1, \frac{f(a_i|\phi_{\theta'}(x_i))}{\mu(a_i|x_i)}\right\}$.
**end for**
$$Q^{ret} \leftarrow \begin{cases} 0 & \text{for terminal } x_k \\ \sum_a Q_{\theta'_v}(x_k, a) f(a|\phi_{\theta'}(x_k)) & \text{otherwise} \end{cases}$$
**for** $i \in \{k-1, \cdots, 0\}$ **do**
 $Q^{ret} \leftarrow r_i + \gamma Q^{ret}$
 $V_i \leftarrow \sum_a Q_{\theta'_v}(x_i, a) f(a|\phi_{\theta'}(x_i))$
 Computing quantities needed for trust region updating:

$$\begin{aligned} g \quad \leftarrow \quad & \min\{c, \rho_i(a_i)\} \nabla_{\phi_{\theta'}(x_i)} \log f(a_i|\phi_{\theta'}(x_i))(Q^{ret} - V_i) \\ & + \sum_a \left[1 - \frac{c}{\rho_i(a)}\right]_+ f(a|\phi_{\theta'}(x_i)) \nabla_{\phi_{\theta'}(x_i)} \log f(a|\phi_{\theta'}(x_i))(Q_{\theta'_v}(x_i, a_i) - V_i) \\ k \quad \leftarrow \quad & \nabla_{\phi_{\theta'}(x_i)} D_{KL}\left[f(\cdot|\phi_{\theta_a}(x_i)) \| f(\cdot|\phi_{\theta'}(x_i))\right] \end{aligned}$$

 Accumulate gradients wrt $\theta'$: $d\theta' \leftarrow d\theta' + \frac{\partial \phi_{\theta'}(x_i)}{\partial \theta'}\left(g - \max\left\{0, \frac{k^T g - \delta}{\|k\|_2^2}\right\}k\right)$
 Accumulate gradients wrt $\theta'_v$: $d\theta_v \leftarrow d\theta_v + \nabla_{\theta'_v}(Q^{ret} - Q_{\theta'_v}(x_i, a))^2$
 Update Retrace target: $Q^{ret} \leftarrow \bar{\rho}_i\left(Q^{ret} - Q_{\theta'_v}(x_i, a_i)\right) + V_i$
**end for**
Perform asynchronous update of $\theta$ using $d\theta$ and of $\theta_v$ using $d\theta_v$.
Updating the average policy network: $\theta_a \leftarrow \alpha\theta_a + (1-\alpha)\theta$

---

# B   $Q(\lambda)$ WITH OFF-POLICY CORRECTIONS

Given a trajectory generated under the behavior policy $\mu$, the $Q(\lambda)$ with off-policy corrections estimator (Harutyunyan et al., 2016) can be expressed recursively as follows:

$$Q^{\text{opc}}(x_t, a_t) = r_t + \gamma[Q^{\text{opc}}(x_{t+1}, a_{t+1}) - Q(x_{t+1}, a_{t+1})] + \gamma V(x_{t+1}). \tag{21}$$

Notice that $Q^{\text{opc}}(x_t, a_t)$ is the same as Retrace with the exception that the truncated importance ratio is replaced with 1.

---

**Algorithm 3** ACER for Continuous Actions

Reset gradients $d\theta \leftarrow 0$ and $d\theta_v \leftarrow 0$.
Initialize parameters $\theta' \leftarrow \theta$ and $\theta'_v \leftarrow \theta_v$.
Sample the trajectory $\{x_0, a_0, r_0, \mu(\cdot|x_0), \cdots, x_k, a_k, r_k, \mu(\cdot|x_k)\}$ from the replay memory.
**for** $i \in \{0, \cdots, k\}$ **do**
 Compute $f(\cdot|\phi_{\theta'}(x_i))$, $V_{\theta'_v}(x_i)$, $\widetilde{Q}_{\theta'_v}(x_i, a_i)$, and $f(\cdot|\phi_{\theta_a}(x_i))$.
 Sample $a'_i \sim f(\cdot|\phi_{\theta'}(x_i))$
 $\rho_i \leftarrow \frac{f(a_i|\phi_{\theta'}(x_i))}{\mu(a_i|x_i)}$ and $\rho'_i \leftarrow \frac{f(a'_i|\phi_{\theta'}(x_i))}{\mu(a'_i|x_i)}$
 $c_i \leftarrow \min\left\{1, (\rho_i)^{\frac{1}{d}}\right\}$.
**end for**
$Q^{ret} \leftarrow \begin{cases} 0 & \text{for terminal } x_k \\ V_{\theta'_v}(x_k) & \text{otherwise} \end{cases}$
$Q^{opc} \leftarrow Q^{ret}$
**for** $i \in \{k-1, \cdots, 0\}$ **do**
 $Q^{ret} \leftarrow r_i + \gamma Q^{ret}$
 $Q^{opc} \leftarrow r_i + \gamma Q^{opc}$
 Computing quantities needed for trust region updating:

$$\begin{aligned} g &\leftarrow \min\{c, \rho_i\} \nabla_{\phi_{\theta'}(x_i)} \log f(a_i|\phi_{\theta'}(x_i)) \left(Q^{opc}(x_i, a_i) - V_{\theta'_v}(x_i)\right) \\ &+ \left[1 - \frac{c}{\rho'_i}\right]_+ (\widetilde{Q}_{\theta'_v}(x_i, a'_i) - V_{\theta'_v}(x_i)) \nabla_{\phi_{\theta'}(x_i)} \log f(a'_i|\phi_{\theta'}(x_i)) \\ k &\leftarrow \nabla_{\phi_{\theta'}(x_i)} D_{KL}\left[f(\cdot|\phi_{\theta_a}(x_i)) \| f(\cdot|\phi_{\theta'}(x_i))\right] \end{aligned}$$

 Accumulate gradients wrt $\theta$: $d\theta \leftarrow d\theta + \frac{\partial \phi_{\theta'}(x_i)}{\partial \theta'}\left(g - \max\left\{0, \frac{k^T g - \delta}{\|k\|_2^2}\right\}k\right)$
 Accumulate gradients wrt $\theta'_v$: $d\theta_v \leftarrow d\theta_v + (Q^{ret} - \widetilde{Q}_{\theta'_v}(x_i, a_i))\nabla_{\theta'_v}\widetilde{Q}_{\theta'_v}(x_i, a_i)$
 $d\theta_v \leftarrow d\theta_v + \min\{1, \rho_i\}\left(Q^{ret}(x_t, a_i) - \widetilde{Q}_{\theta'_v}(x_t, a_i)\right)\nabla_{\theta'_v}V_{\theta'_v}(x_i)$
 Update Retrace target: $Q^{ret} \leftarrow c_i\left(Q^{ret} - \widetilde{Q}_{\theta'_v}(x_i, a_i)\right) + V_{\theta'_v}(x_i)$
 Update Retrace target: $Q^{opc} \leftarrow \left(Q^{opc} - \widetilde{Q}_{\theta'_v}(x_i, a_i)\right) + V_{\theta'_v}(x_i)$
**end for**
Perform asynchronous update of $\theta$ using $d\theta$ and of $\theta_v$ using $d\theta_v$.
Updating the average policy network: $\theta_a \leftarrow \alpha\theta_a + (1-\alpha)\theta$

---

Because of the lack of the truncated importance ratio, the operator defined by $Q^{\text{opc}}$ is only a contraction if the target and behavior policies are close to each other (Harutyunyan et al., 2016). $Q(\lambda)$ with off-policy corrections is therefore less stable compared to Retrace and unsafe for policy evaluation.

$Q^{\text{opc}}$, however, could better utilize the returns as the traces are not cut by the truncated importance weights. As a result, $Q^{\text{opc}}$ could be used efficiently to estimate $Q^\pi$ in policy gradient (e.g. in Equation (16)). In our continuous control experiments, we have found that $Q^{\text{opc}}$ leads to faster learning.

## C   Retrace as Truncated Importance Sampling with Bias Correction

For the purpose of proving proposition 1, we assume our environment to be a Markov Decision Process $(\mathcal{X}, \mathcal{A}, \gamma, P, r)$. We restrict $\mathcal{X}$ to be a finite state space. For notational simplicity, we also restrict $\mathcal{A}$ to be a finite action space. $P : \mathcal{X}, \mathcal{A} \to \mathcal{X}$ defines the state transition probabilities and $r : \mathcal{X}, \mathcal{A} \to [-R_{\text{MAX}}, R_{\text{MAX}}]$ defines a reward function. Finally, $\gamma \in [0, 1)$ is the discount factor.

*Proof of proposition 1.* First we show that $\mathcal{B}$ is a contraction operator.

$$|\mathcal{B}Q(x,a) - Q^\pi(x,a)|$$

$$= \left| \mathbb{E}_\mu \left[ \sum_{t \geq 0} \gamma^t \left( \prod_{i=1}^t \bar{\rho}_i \right) \left( \gamma \mathop{\mathbb{E}}_{b \sim \pi} \left( \left[ \frac{\rho_{t+1}(b) - c}{\rho_{t+1}(b)} \right]_+ (Q(x_{t+1}, b) - Q^\pi(x_{t+1}, b)) \right) \right) \right] \right|$$

$$\leq \mathbb{E}_\mu \left[ \sum_{t \geq 0} \gamma^t \left( \prod_{i=1}^t \bar{\rho}_i \right) \left[ \gamma \mathop{\mathbb{E}}_{b \sim \pi} \left( \left[ \frac{\rho_{t+1}(b) - c}{\rho_{t+1}(b)} \right]_+ |Q(x_{t+1}, b) - Q^\pi(x_{t+1}, b)| \right) \right] \right]$$

$$\leq \mathbb{E}_\mu \left[ \sum_{t \geq 0} \gamma^t \left( \prod_{i=1}^t \bar{\rho}_i \right) \left( \gamma(1 - \bar{\mathrm{P}}_{t+1}) \sup_b |Q(x_{t+1}, b) - Q^\pi(x_{t+1}, b)| \right) \right] \qquad (22)$$

Where $\bar{\mathrm{P}}_{t+1} = 1 - \mathop{\mathbb{E}}_{b \sim \pi} \left[ \left[ \frac{\rho_{t+1}(b) - c}{\rho_{t+1}(b)} \right]_+ \right] = \mathop{\mathbb{E}}_{b \sim \mu} [\bar{\rho}_{t+1}(b)]$. The last inequality in the above equation is due to Hölder's inequality.

$$(22) \leq \sup_{x,b} |Q(x,b) - Q^\pi(x,b)| \mathbb{E}_\mu \left[ \sum_{t \geq 0} \gamma^t \left( \prod_{i=1}^t \bar{\rho}_i \right) (\gamma(1 - \bar{\mathrm{P}}_{t+1})) \right]$$

$$= \sup_{x,b} |Q(x,b) - Q^\pi(x,b)| \mathbb{E}_\mu \left[ \gamma \sum_{t \geq 0} \gamma^t \left( \prod_{i=1}^t \bar{\rho}_i \right) - \sum_{t \geq 0} \gamma^t \left( \prod_{i=1}^t \bar{\rho}_i \right) (\gamma \bar{\mathrm{P}}_{t+1}) \right]$$

$$= \sup_{x,b} |Q(x,b) - Q^\pi(x,b)| \mathbb{E}_\mu \left[ \gamma \sum_{t \geq 0} \gamma^t \left( \prod_{i=1}^t \bar{\rho}_i \right) - \sum_{t \geq 0} \gamma^{t+1} \left( \prod_{i=1}^{t+1} \bar{\rho}_i \right) \right]$$

$$= \sup_{x,b} |Q(x,b) - Q^\pi(x,b)| (\gamma C - (C - 1))$$

where $C = \sum_{t \geq 0} \gamma^t \left( \prod_{i=1}^t \bar{\rho}_i \right)$. Since $C \geq \sum_{t=0}^0 \gamma^t \left( \prod_{i=1}^t \bar{\rho}_i \right) = 1$, we have that $\gamma C - (C - 1) \leq \gamma$. Therefore, we have shown that $\mathcal{B}$ is a contraction operator.

Now we show that $\mathcal{B}$ is the same as Retrace. By apply the trunction and bias correction trick, we have

$$\mathop{\mathbb{E}}_{b \sim \pi} [Q(x_{t+1}, b)] = \mathop{\mathbb{E}}_{b \sim \mu} [\bar{\rho}_{t+1}(b) Q(x_{t+1}, b)] + \mathop{\mathbb{E}}_{b \sim \pi} \left( \left[ \frac{\rho_{t+1}(b) - c}{\rho_{t+1}(b)} \right]_+ Q(x_{t+1}, b) \right). \qquad (23)$$

By adding and subtracting the two sides of Equation (23) inside the summand of Equation (20), we have

$$\mathcal{B}Q(x,a) = \mathbb{E}_\mu \left[ \sum_{t \geq 0} \gamma^t \left( \prod_{i=1}^t \bar{\rho}_i \right) \left[ r_t + \gamma \mathop{\mathbb{E}}_{b \sim \pi} \left( \left[ \frac{\rho_{t+1}(b) - c}{\rho_{t+1}(b)} \right]_+ Q(x_{t+1}, b) \right) + \gamma \mathop{\mathbb{E}}_{b \sim \pi} [Q(x_{t+1}, b)] \right. \right.$$

$$\left. \left. - \gamma \mathop{\mathbb{E}}_{b \sim \mu} [\bar{\rho}_{t+1}(b) Q(x_{t+1}, b)] - \gamma \mathop{\mathbb{E}}_{b \sim \pi} \left( \left[ \frac{\rho_{t+1}(b) - c}{\rho_{t+1}(b)} \right]_+ Q(x_{t+1}, b) \right) \right] \right]$$

$$= \mathbb{E}_\mu \left[ \sum_{t \geq 0} \gamma^t \left( \prod_{i=1}^t \bar{\rho}_i \right) \left( r_t + \gamma \mathop{\mathbb{E}}_{b \sim \pi} [Q(x_{t+1}, b)] - \gamma \mathop{\mathbb{E}}_{b \sim \mu} [\bar{\rho}_{t+1}(b) Q(x_{t+1}, b)] \right) \right]$$

$$= \mathbb{E}_\mu \left[ \sum_{t \geq 0} \gamma^t \left( \prod_{i=1}^t \bar{\rho}_i \right) \left( r_t + \gamma \mathop{\mathbb{E}}_{b \sim \pi} [Q(x_{t+1}, b)] - \gamma \bar{\rho}_{t+1} Q(x_{t+1}, a_{t+1}) \right) \right]$$

$$= \mathbb{E}_\mu \left[ \sum_{t \geq 0} \gamma^t \left( \prod_{i=1}^t \bar{\rho}_i \right) \left( r_t + \gamma \mathop{\mathbb{E}}_{b \sim \pi} [Q(x_{t+1}, b)] - Q(x_t, a_t) \right) \right] + Q(x,a) = \mathcal{R}Q(x,a)$$

□

In the remainder of this appendix, we show that $\mathcal{B}$ generalizes both the Bellman operator and importance sampling. First, we reproduce the definition of $\mathcal{B}$:

$$\mathcal{B}Q(x,a) = \mathbb{E}_\mu \left[ \sum_{t \geq 0} \gamma^t \left( \prod_{i=1}^{t} \bar{\rho}_i \right) \left( r_t + \gamma \underset{b \sim \pi}{\mathbb{E}} \left( \left[ \frac{\rho_{t+1}(b) - c}{\rho_{t+1}(b)} \right]_+ Q(x_{t+1}, b) \right) \right) \right].$$

When $c = 0$, we have that $\bar{\rho}_i = 0 \ \forall i$. Therefore only the first summand of the sum remains:

$$\mathcal{B}Q(x,a) = \mathbb{E}_\mu \left[ r_t + \gamma \underset{b \sim \pi}{\mathbb{E}} \left( Q(x_{t+1}, b) \right) \right].$$

In this case $\mathcal{B} = \mathcal{T}$. When $c = \infty$, the compensation term disappears and $\bar{\rho}_i = \rho_i \ \forall i$:

$$\mathcal{B}Q(x,a) = \mathbb{E}_\mu \left[ \sum_{t \geq 0} \gamma^t \left( \prod_{i=1}^{t} \rho_i \right) \left( r_t + \gamma \underset{b \sim \pi}{\mathbb{E}} \left( 0 \times Q(x_{t+1}, b) \right) \right) \right] = \mathbb{E}_\mu \left[ \sum_{t \geq 0} \gamma^t \left( \prod_{i=1}^{t} \rho_i \right) r_t \right].$$

In this case $\mathcal{B}$ is the same operator defined by importance sampling.

## D    DERIVATION OF $V^{target}$

By using the truncation and bias correction trick, we can derive the following:

$$V^\pi(x_t) = \underset{a \sim \mu}{\mathbb{E}} \left[ \min \left\{ 1, \frac{\pi(a|x_t)}{\mu(a|x_t)} \right\} Q^\pi(x_t, a) \right] + \underset{a \sim \pi}{\mathbb{E}} \left( \left[ \frac{\rho_t(a) - 1}{\rho_t(a)} \right]_+ Q^\pi(x_{t+1}, a) \right).$$

We, however, cannot use the above equation as a target as we do not have access to $Q^\pi$. To derive a target, we can take a Monte Carlo approximation of the first expectation in the RHS of the above equation and replace the first occurrence of $Q^\pi$ with $Q^{ret}$ and the second with our current neural net approximation $Q_{\theta_v}(x_t, \cdot)$:

$$V_{pre}^{target}(x_t) := \min \left\{ 1, \frac{\pi(a_t|x_t)}{\mu(a_t|x_t)} \right\} Q^{ret}(x_t, a_t) + \underset{a \sim \pi}{\mathbb{E}} \left( \left[ \frac{\rho_t(a) - 1}{\rho_t(a)} \right]_+ Q_{\theta_v}(x_t, a) \right). \quad (24)$$

Through the truncation and bias correction trick again, we have the following identity:

$$\underset{a \sim \pi}{\mathbb{E}} [Q_{\theta_v}(x_t, a)] = \underset{a \sim \mu}{\mathbb{E}} \left[ \min \left\{ 1, \frac{\pi(a|x_t)}{\mu(a|x_t)} \right\} Q_{\theta_v}(x_t, a) \right] + \underset{a \sim \pi}{\mathbb{E}} \left( \left[ \frac{\rho_t(a) - 1}{\rho_t(a)} \right]_+ Q_{\theta_v}(x_t, a) \right). \quad (25)$$

Adding and subtracting both sides of Equation (25) to the RHS of (24) while taking a Monte Carlo approximation, we arrive at $V^{target}(x_t)$:

$$V^{target}(x_t) := \min \left\{ 1, \frac{\pi(a_t|x_t)}{\mu(a_t|x_t)} \right\} \left( Q^{ret}(x_t, a_t) - Q_{\theta_v}(x_t, a_t) \right) + V_{\theta_v}(x_t).$$

## E    CONTINUOUS CONTROL EXPERIMENTS

### E.1    DESCRIPTION OF THE CONTINUOUS CONTROL PROBLEMS

Our continuous control tasks were simulated using the MuJoCo physics engine (Todorov et al. (2012)). For all experiments we considered an episodic setup with an episode length of $T = 500$ steps and a discount factor of 0.99.

**Cartpole swingup**    This is an instance of the classic cart-pole swing-up task. It consists of a pole attached to a cart running on a finite track. The agent is required to balance the pole near the center of the track by applying a force to the cart only. An episode starts with the pole at a random angle and zero velocity. A reward zero is given except when the pole is approximately upright (within $\pm 5 \deg$) and the track approximately in the center of the track ($\pm 0.05$) for a track length of 2.4. The observations include position and velocity of the cart, angle and angular velocity of the pole. a sine/cosine of the angle, the position of the tip of the pole, and Cartesian velocities of the pole. The dimension of the action space is 1.

**Reacher3** The agent needs to control a planar 3-link robotic arm in order to minimize the distance between the end effector of the arm and a target. Both arm and target position are chosen randomly at the beginning of each episode. The reward is zero except when the tip of the arm is within 0.05 of the target, where it is one. The 8-dimensional observation consists of the angles and angular velocity of all joints as well as the displacement between target and the end effector of the arm. The 3-dimensional action are the torques applied to the joints.

**Cheetah** The Half-Cheetah (Wawrzyński (2009); Heess et al. (2015)) is a planar locomotion task where the agent is required to control a 9-DoF cheetah-like body (in the vertical plane) to move in the direction of the x-axis as quickly as possible. The reward is given by the velocity along the x-axis and a control cost: $r = v_x + 0.1\|a\|^2$. The observation vector consists of the z-position of the torso and its $x, z$ velocities as well as the joint angles and angular velocities. The action dimension is 6.

**Fish** The goal of this task is to control a 13-DoF fish-like body to swim to a random target in 3D space. The reward is given by the distance between the head of the fish and the target, a small penalty for the body not being upright, and a control cost. At the beginning of an episode the fish is initialized facing in a random direction relative to the target. The 24-dimensional observation is given by the displacement between the fish and the target projected onto the torso coordinate frame, the joint angles and velocities, the cosine of the angle between the z-axis of the torso and the world z-axis, and the velocities of the torso in the torso coordinate frame. The 5-dimensional actions control the position of the side fins and the tail.

**Walker** The 9-DoF planar walker is inspired by (Schulman et al. (2015a)) and is required to move forward along the x-axis as quickly as possible without falling. The reward consists of the x-velocity of the torso, a quadratic control cost, and terms that penalize deviations of the torso from the preferred height and orientation (i.e. terms that encourage the walker to stay standing and upright). The 24-dimensional observation includes the torso height, velocities of all DoFs, as well as sines and cosines of all body orientations in the x-z plane. The 6-dimensional action controls the torques applied at the joints. Episodes are terminated early with a negative reward when the torso exceeds upper and lower limits on its height and orientation.

**Humanoid** The humanoid is a 27 degrees-of-freedom body with 21 actuators (21 action dimensions). It is initialized lying on the ground in a random configuration and the task requires it to achieve a standing position. The reward function penalizes deviations from the height of the head when standing, and includes additional terms that encourage upright standing, as well as a quadratic action penalty. The 94 dimensional observation contains information about joint angles and velocities and several derived features reflecting the body's pose.

E.2    UPDATE EQUATIONS OF THE BASELINE TIS

The baseline TIS follows the following update equations,

$$\text{updates to the policy: } \min\left\{5, \left(\prod_{i=0}^{k-1}\rho_{t+i}\right)\right\}\left[\sum_{i=0}^{k-1}\gamma^i r_{t+i} + \gamma^k V_{\theta_v}(x_{k+t}) - V_{\theta_v}(x_t)\right]\nabla_\theta \log \pi_\theta(a_t|x_t),$$

$$\text{updates to the value: } \min\left\{5, \left(\prod_{i=0}^{k-1}\rho_{t+i}\right)\right\}\left[\sum_{i=0}^{k-1}\gamma^i r_{t+i} + \gamma^k V_{\theta_v}(x_{k+t}) - V_{\theta_v}(x_t)\right]\nabla_{\theta_v} V_{\theta_v}(x_t).$$

The baseline Trust-TIS is appropriately modified according to the trust region update described in Section 3.3.

E.3    SENSITIVITY ANALYSIS

In this section, we assess the sensitivity of ACER to hyper-parameters. In Figures 5 and 6, we show, for each game, the final performance of our ACER agent versus the choice of learning rates, and the trust region constraint $\delta$ respectively.

Note, as we are doing random hyper-parameter search, each learning rate is associated with a random $\delta$ and vice versa. It is therefore difficult to tease out the effect of either hyper-parameter independently.

We observe, however, that ACER is not very sensitive to the hyper-parameters overall. In addition, smaller $\delta$'s do not seem to adversely affect the final performance while larger $\delta$'s do in domains of higher action dimensionality. Similarly, smaller learning rates perform well while bigger learning rates tend to hurt final performance in domains of higher action dimensionality.

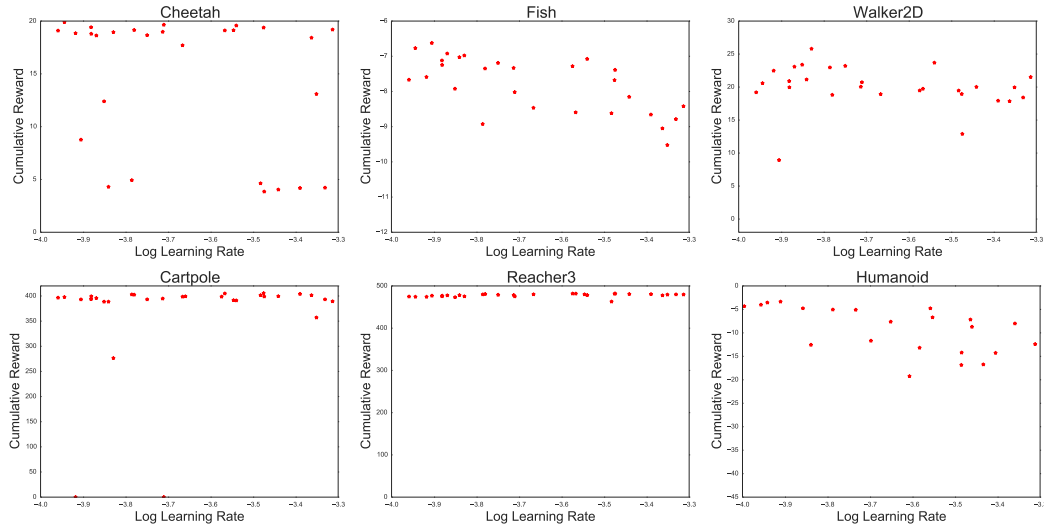

Figure 5: Log learning rate vs. cumulative rewards in all the continuous control tasks for ACER. The plots show the final performance after training for all 30 log learning rates considered. Note that each learning rate is associated with a different $\delta$ as a consequence of random search over hyper-parameters.

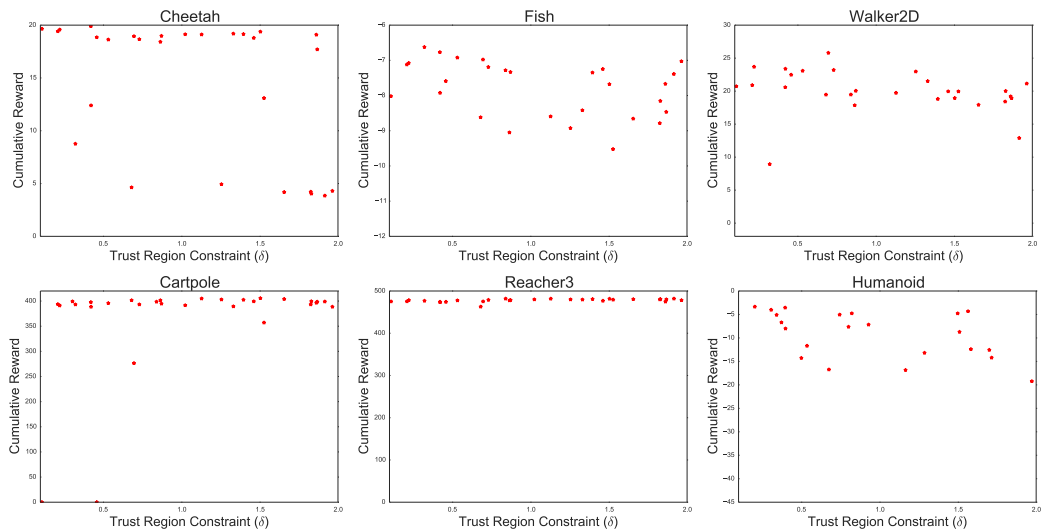

Figure 6: Trust region constraint ($\delta$) vs. cumulative rewards in all the continuous control tasks for ACER. The plots show the final performance after training for all 30 trust region constraints ($\delta$) searched over. Note that each $\delta$ is associated with a different learning rate as a consequence of random search over hyper-parameters.

### E.4 EXPERIMENTAL SETUP OF ABLATION ANALYSIS

For the ablation analysis, we use the same experimental setup as in the continuous control experiments while removing one component at a time.

To evaluate the effectiveness of Retrace/Q($\lambda$) with off-policy correction, we replace both with importance sampling based estimates (following Degris et al. (2012)) which can be expressed recursively: $R_t = r_t + \rho_{t+1} R_{t+1}$.

To evaluate the Stochastic Dueling Networks, we replace it with two separate networks: one computing the state values and the other $Q$ values. Given $Q^{ret}(x_t, a_t)$, the naive way of estimating the state values is to use the following update rule:

$$\left(\rho_t Q^{ret}(x_t, a_t) - V_{\theta_v}(x_t)\right) \nabla_{\theta_v} V_{\theta_v}(x_t).$$

The above update rule, however, suffers from high variance. We consider instead the following update rule:

$$\rho_t \left(Q^{ret}(x_t, a_t) - V_{\theta_v}(x_t)\right) \nabla_{\theta_v} V_{\theta_v}(x_t)$$

which has markedly lower variance. We update our $Q$ estimates as before.

To evaluate the effects of the truncation and bias correction trick, we change our $c$ parameter (see Equation (16)) to $\infty$ so as to use pure importance sampling.

