# Peer review of "Sample Efficient Actor-Critic with  Experience Replay"

_ICLR 2017 — accepted_

[Public Comment · Tara N Sainath · 07 Nov 2016]
**ICLR Paper Format**

Dear Authors,

Please resubmit your paper in the ICLR 2017 format with the correct marging spacing for your submission to be considered. Thank you!

[Public Comment · (anonymous) · 14 Dec 2016]
**clarification of Eq 3**

Should the inside summations of equation (3) should go from i = 0 to (k - t)?

[Official Review · AnonReviewer3 · rating 6 · confidence 4 · 15 Dec 2016]

The paper looks at several innovations for deep RL, and evaluates their effect on solving games in the Atari domain.  The paper reads a bit like a laundry list of the researcher’s latest tricks.  It is written clearly enough, but lacks a compelling message.  I expect the work will be interesting to people already implementing deep RL methods, but will probably not get much attention from the broader community.

The claims on p.1 suggest the approach is stable and sample efficience, and so I expected to see some theoretical analysis with respect to these properties. But this is an empirical claim; it would help to clarify that in the abstract.

The proposed innovations are based on sound methods.  It is particularly nice to see the same approach working for both discrete and continuous domains.

The paper has reasonably complete empirical results. It would be nice to see confidence intervals on more of the plots. Also, the results don’t really tease apart the effect of each of the various innovations, so it’s harder to understand the impact of each piece and to really get intuition, for example about why ACER outperforms A3C.  Also, it wasn’t clear to me why you only get matching results on discrete tasks, but get state-of-the-art on continuous tasks.

The paper has good coverage of the related literature. It is nice to see this work draw more attention to Retrace, including the theoretical characterization in Sec.7.

[Official Review · AnonReviewer2 · rating 6 · confidence 3 · 16 Dec 2016]

This paper introduces an actor-critic deep RL approach with experience replay, which combines truncated importance sampling and trust region policy optimization. The paper also proposes a new method called stochastic duelling networks to estimate the critic for continuous action spaces. The method is applied to Atari games and continuous control problems, where it yields performance comparable to state-of-the-art methods.

As mentioned in the beginning of the paper, the main contributions of this work lies in combining 1) truncated importance sampling with retrace, 2) trust region policy optimization, and 3) stochastic duelling networks. These improvements work well and may be beneficial to future work in RL.

However, each improvement appears to be quite incremental. Moreover, the ACER framework seems much more complex and fragile to implement compared to the standard deep q-learning with prioritized replay (which appears to perform just as well on Atari games). So for the Atari domain, I would still put my money on prioritized replay due to its simplicity. Thirdly, improving sample efficiency for deep RL is a laudable goal, but really this goal should be pursued in a problem setting where sample efficiency is important. Unfortunately, the paper only evaluates sample efficiency in the Atari and continuous control tasks domain; two domains where sample efficiency is not important. Thus, it is not clear that the proposed method ACER will generalize to problems where we really care about sample efficiency.

Some technical aspects which need clarifications:
- For Retrace, I assume that you compute recursively $Q^{ret}$ starting from the end of each trajectory? Please comment on this.
- It's not clear to me how to derive eq. (7). Is an approximation (double tilde) sign missing?
- In section 3.1 the paper argued that $Q^{ret}$ gives a lower-variance estimate of the action-value function. Then why not use it in eq. (8) for the bias correction term?
- The paper states that it uses a replay memory of 50000 frames, so that across threads it is comparable in size to previous work. However, for each thread this is much smaller compared to earlier experiments on Atari games. For example, one million experience replay transitions were used in the paper "Prioritized Experience Replay" by Schaul et al. This may have a huge impact on performance of the models (both for ACER and for the competing models). In order to properly assess the improvements of ACER over previous work, the authors need to also experiment with larger experience replay memories.


Other comments:
- Please move Section 7 to the appendix.
- "Moreover, when using small values of lambda to reduce variance, occasional large importance weights can still cause instability": I think what is meant is using *large* values of lambda.
- Above eq. (6) mention that the squared error is used.
- Missing a "t" subscript at the beginning of eq. (9)?
- It was hard to understand the stochastic duelling networks. Please rephrase this part.
- Please clarify this sentence "To compare different agents, we adopt as our metric the median of the human normalized score over all 57 games."
- Figure 2 (Bottom): Please add label to vertical axes.

[Official Review · AnonReviewer1 · rating 7 · confidence 3 · 25 Dec 2016 (modified: 21 Jan 2017)]
**No Title**

This paper studies the off-policy learning of actor-critic with experience replay. This is an important and challenging problem in order to improve the sample efficiency of the reinforcement learning algorithms. The paper attacks the problem by introducing a new way to truncate importance weight, a modified trust region optimization, and by combining retrace method. The combination of the above techniques performs well on Atari and MuJoCo in terms of improving sample efficiency. My main comment is how does each of the technique contribute to the performance gain? If some experiments could be carried out to evaluate the separate gains from these tricks, it would be helpful.

[Author Response · ziyu wang · 14 Jan 2017]
**Ablations and why each ingredient is an important contribution on its own**

We thank the three reviewers. The one common concern is ablations. This paper proposes several new ideas, and then goes on to combine these ideas.  

To answer the reviewers concerns about ablations, we added a new figure (Figure 4). This is an extremely important figure and we urge the reviewers and readers to consult it as it should answer any concerns and highlight the value of the many contributions made in this paper. The figure shows that each ingredient (Retrace/Q-lambda with off-policy correction, stochastic dueling nets, and the NEW trust region method) on its own leads to a massive improvement. Likewise, truncation with bias correction plays an important role for large action spaces (control of humanoid). This figure indicates that this paper is not about making 4 small contributions and combining them. Rather it is about making 4 important contributions, which are all essential to obtain a stable, scalable, general, off-policy actor critic. Attaining this has been a holy grail, and this paper shows how to do it.

Given our good results, we could easily have written several papers; one for each contribution. Instead, we chose to do the honest thing and write a single solid 20-page paper aimed at truly building powerful deep RL agents for both continuous and discrete action spaces. The paper also presents novel theoretical results for RL, and a very comprehensive experimental study. 

We did not want to claim state-of-the-art on Atari because this often depends on how one chooses to measure what should be state-of-the-art (eg sample complexity, highest median, highest mean, etc.). But clearly, in terms of median, ACER with 1 replay achieves a higher median score that any previously reported result. Note that this result is not just for a few games, but for the entire set of 57 games. The UNREAL agent submitted to this conference is the only method we know that achieves a higher median, but it does so by adding auxiliary tasks to A3C and massive hyper-parameter sweeps. We could also add auxiliary tasks to ACER and do hyper-parameter sweeps to further improve it, but this is left for future work as we wanted to focus on designing a powerful core RL agent.

We hope this reply and in particular the ablations clearly answer your concerns. With 3 6’s this thorough paper will be rejected despite the several novel contributions it makes, new theoretical analysis, and excellent results on a comprehensive set of tasks. We hope you take the ablations and this reply into consideration to choose your final scores.

[Author Response · ziyu wang · 21 Jan 2017]
**General Comments**

Dear reviewers, we would really appreciate it if you can take a look at the paper again in light of our replies, the updated paper, and the comments from Xi Chen. Thanks very much for your time!

[Public Comment · Xi Chen · 21 Jan 2017]
**Important contributions**

This submission has a couple important contributions and it'd be actually easy to split it into 2 strong papers.

Roughly:
1. Especially in deep rl, policy gradient methods have suffered from worse sample complexity compared to value-based methods like DQN. Learning a critic to improve sample efficiency for policy gradient methods is a straightforward idea but this is the first convincing demonstration (by carefully combing different elements like Retrace(\lambda) and experience replay). This represents an important step towards making policy gradient methods more sample efficient and alone, I believe, merits acceptance. It's worth noting that there is another ICLR submission Q-Prop (

[Public Comment · (anonymous) · 20 Apr 2017]
**Equation 4**

First of all, thanks for this excellent work.

My question is about eq. 4. In Degris et al (2012) the policy gradient is computed as the expectation under the off-policy behavior of \rho(s_t, a_t) \psi(s_t, a_t) (R_t^\lambda - V(s_t))
With \rho(s_t,a_t) = \pi(a_t | s_t) / \mu(a_t | s_t) and \psi(s_t, a_t) = \grad_\theta ( log \pi (a_t | s_t) ) /  \pi (a_t | s_t)
The last division by \pi (a_t | s_t) is missing in equation (4).

Am I mistaken or is the reference wrong?
Thanks for your time.

[Final Decision · Program Chairs · 06 Feb 2017]
**ICLR committee final decision**

pros:
 - set of contributions leading to SOTA for sample complexity wrt Atari (discrete) and continuous domain problems
 - significant experimental analysis
 - long all-in-one paper
 
 cons:
 - builds on existing ideas, although ablation analysis shows each to be essential
 - long paper
 
The PCs believe this paper will be a good contribution to the conference track.